# OpenReview forum: "Proving Olympiad Algebraic Inequalities without Human Demonstrations"
_NeurIPS.cc/2024/Workshop/MATH-AI — MATH-AI 24_

### Official Review · Reviewer_Xp6b · 2024-09-27
**A Good Work that combines value networks with Sympy to prove IMO-level inequalties**

**Rating:** 7
**Confidence:** 5

**Review:**

This paper introduces AIPS, an approach for proving IMO-level inequalities. It first proposes automatically generating a large-scale dataset that contains complex cyclic symmetric inequalities by sampling from a predefined set of inequality theorems. A model is then fine-tuned on this dataset to serve as a value network, evaluating the difficulty of inequality expressions. This value network is combined with SymPy's symbolic deductive engine to perform a proof search. Experiments demonstrate that AIPS significantly outperforms a range of existing baselines, including pure LLMs and neural theorem provers, on 20 challenging IMO inequalities.

The paper is overall easy to read. While the high-level idea of this work is similar to prior work, the focus on inequality proving—a relatively underexplored domain in automated theorem proving—is a valuable contribution. The ability of AIPS to achieve 50% accuracy on IMO-level inequalities is noteworthy, given the complexity of these problems.

That said, the paper would benefit from structural improvements. The introduction is overly long, taking up too much space, while the methodology section is comparatively brief and lacks depth in some key areas. Shifting some of the content from the appendix to the main body, particularly details on the dataset generation process and the technical specifics of the value network, would provide a more balanced and informative presentation.

---

### Official Review · Reviewer_cwxV · 2024-10-06
**Could potentially be quite big for theorem proving dataset creation**

**Rating:** 8
**Confidence:** 3

**Review:**

### Summary

The paper introduces **AIPS**, a model that autonomously generates and proves complex inequality theorems at the IMO level without human demonstrations. The system uses a symbolic deductive combined with a value network trained through curriculum learning to guide proof search efficiently. AIPS proves inequality theorems and outperforms state-of-the-art systems, solving 10 out of 20 Olympiad-level problems.

### Strengths

1. AIPS's ability to autonomously generate and prove Olympiad-level inequality theorems without human input is a major advancement in automated reasoning. This opens up possibilities for generating vast amounts of training data for theorem-proving systems.
2. The use of a value network trained through a curriculum learning strategy enables AIPS to improve its performance progressively, demonstrating efficient problem-solving strategies akin to human intuition. This method allows for better scalability and adaptability across problems of varying difficulty.
3. The paper demonstrates that AIPS outperforms previous state-of-the-art systems, solving 10 out of 20 IMO-level problems.
4. The system's ability to generate theorems that were used in real-world Olympiad competitions shows its potential for contributing novel and challenging mathematical problems for human mathematicians.
5. The paper provides a comprehensive evaluation of AIPS across various test sets and baselines, including comparisons with large language models (LLMs) and other formal theorem provers.

### Weaknesses

1. While AIPS excels in algebraic inequalities, the scope of the system is limited to this domain. It remains unclear how the system would generalize to other types of Olympiad-level mathematical problems, such as those involving geometry, number theory, or combinatorics.
2. The reliance on SymPy for symbolic manipulation introduces a potential bottleneck for more complex or non-standard inequalities, which may require specialized transformation rules beyond what SymPy currently supports.
3. Although the paper claims strong results, it does not provide a detailed breakdown of the computational resources required to run AIPS at scale. Given the complexity of proof searches, the system might face limitations in terms of computational efficiency when dealing with larger or more complex problem sets.
4. While AIPS outperforms previous models, the paper lacks a detailed analysis of its failure cases. Understanding where and why AIPS fails could help improve future iterations of the system and provide insights into its limitations.

### Questions

1. How would AIPS perform when applied to other mathematical domains beyond algebraic inequalities? Could its methodology be extended to handle problems from geometry or combinatorics?
2. What are the computational costs involved in running AIPS, especially on larger datasets or more complex problem sets? Are there ways to optimize the system for better efficiency?
3. How does AIPS handle ambiguities or non-standard forms of inequalities, particularly in real-world Olympiad problems that may not fit neatly into its predefined transformation rules?
4. Could the system be integrated with interactive theorem provers for further refinement of proofs, especially in cases where the generated proof may require additional human insight?

---

### Official Review · Reviewer_HiZC · 2024-10-07

**Rating:** 7
**Confidence:** 4

**Review:**

Summary: Much like the Euclidean geometry domain, the inequalities domain is also highly constrained and serves as a natural followup domain for an AlphaGeometry-style approach. The authors introduce AIPS, a collection of a symbolic deduction engine and a value network that, when looped in search, is capable of producing proofs to a fair number of olympiad inequality problems.

Significance: The work is significant; it seems to be a natural followup to AlphaGeometry. Similarly, the deductive engine is capable of most of the reasoning but the search tree must be constrained enough to reasonably find a proof. The value network operates somewhat similarly to the auxiliary-construction model from AG in this regard. The overall combined effort can solve some, but not all, inequality problems from popular olympiads. The synthetic problem generator appears to be quite valuable, especially as rated by professional contestants.

I think this would be a nice work for the MATH-AI community to see.

Q: How are the proofs checked? I may have missed this detail, but which proof assistant actually performs the checking? Is it a custom one for inequalities alone?

---

### Official Review · Reviewer_R2Vj · 2024-10-07
**Review of "Proving Olympiad Algebraic Inequalities without Human Demonstrations"**

**Rating:** 8
**Confidence:** 3

**Review:**

Summary
* This work proposes a system for both generating complex inequality theorems (some up to IMO level) and solving Olympiad-level inequality problems without human demonstrations
* The system incorporates a value curriculum learning strategy on the generated datasets to improve performance

Strengths
* Results on both data generation and model performance are strong (generating a problem selected for a math olympiad and outperforming state-of-the-art methods in solving the problems)
* The value curriculum learning results show the novelty and importance of this approach
* Paper is clearly written and concise

Weaknesses
* Missing analysis of the limitations of AIPS, especially an analysis of the proofs that AIPS got incorrect. How close were the solutions produced by AIPS on the 10 problems it got wrong?
* It’s unclear to me whether the filtered synthetic data was only used for the AIPS results. If so, is it possible that fine tuning LLMs on this dataset could improve their performance? An interesting avenue for future work would be to explore how the synthetic data affects the performance of other model categories now that you have the ability to generate data more easily
* Minor errors: typo in Figure 5 caption, repeated sentences in B.2 paragraphs 1 and 2

---

### Decision · Program_Chairs · 2024-10-07

Accept